# STRONG DENOISING OF FINANCIAL TIME-SERIES

## ABSTRACT

In this paper we introduce a method for improving the signal to noise ratio of financial data. The approach relies on combining a target variable with different context variables and using auto-encoders (AEs) to learn reconstructions of the combined inputs. The idea is to seek agreement among multiple AEs which are trained on related but different inputs for which they are forced to find common ground. The training process is set up as a conversation where models take turns at producing a prediction (speaking) or reconciling own predictions with the output of the other AE (listening), until an agreement is reached. This leads to "mutual regularization" among the AEs. Unlike standard regularization which relies on including a complexity penalty into the loss function, the proposed method uses the partner network to detect and amend the lack of generality in the data representation. As only true regularities can be agreed upon by the AEs, the replication of noise is costly and will therefore be avoided.

## 1 INTRODUCTION

Financial data is characterized by an extremely poor signal-to-noise ration and hence poor predictability AQR (2024). Even while some logical explanation of market moves may be given ex-post, future prices appear to be largely random especially when considering short (¡ one year) investment horizons. Financial markets are social systems in which multiple mechanisms continuously compete for the attention of investors. It is common to think of the market as operating in different "regimes" each having its own set of rules on how real-world observations (news) are converted into prices. Auto-encoders (AEs) seem to be well suited to extract and represent these rules. Their design is guided by the manifold hypothesis which states that many datasets of practical interest concentrate around a low-dimensional manifold Meilă & Zhang (2023). The geometry of the manifold reflects structural constraints obeyed by the data which, in turn, are due to some mechanism that prepares the data. AEs project inputs onto the manifold in the encoding stage and "lift" points on the manifold back into the high–dimensional input space during the decoding stage.

The key is to make the encoder as insensitive to variations of the input as possible. Only variations that give rise to a new input pattern (instead of being a noisy version of a given pattern) must be encoded. These variations define the tangent directions of the manifold. Learning is unsupervised so the difference between signal and noise is not known but determined by the auto-encoder or, more precisely, its (in-)sensitivity to inputs. Insensitivity to input data is typically achieved by placing a bottleneck layer at the end of the encoding stage such that the code dimension is lower than the input dimension (undercomplete auto-encoder). Alternatively (or in addition), a regularization term is added to the loss function which limits the complexity of the code e.g., by encouraging zeros in the encoding layer. Yet another method pursues local insensitivity to input variations by penalizing the derivative of the encoding function with respect to inputs Rifai et al. (2011). None of these methods incorporate an explicit statement of the generalization capability of the model. They all approach the problem indirectly by preventing the auto-encoder to represent arbitrary data sets.

### COMMON UNDERSTANDING REQUIRES GENERALITY

In this paper we approach the regularization problem in a different way. Every training step is followed by a reconciliation step in which two networks try and understand each other. The idea is that common understanding can only be obtained if the acquired knowledge (about the data) is general enough. If two AEs disagree on the representation of data they will be forced to alter the

way they encode the data so that different aspects are extracted on which an agreement is possible. The agreement is measured in terms of the recurrence of combined encoding patterns: two networks are said to *agree* if a given encoding keeps appearing in conjunction with the same encoding by the partner AE. The idea is that this only happens if both networks extract *essential* features from the input data which itself keep reappearing. If spurious features are encoded then the probability of recurrence of a specific pair of encodings is low. This means that the agreement level (AL) quantifies the generality of the representation.

The principal idea in this paper is to have two networks create two differing vantage points from which the data is examined. This is realized by a) choosing heterogeneous architectures i.e., networks that differ in the number and width of layers or activation functions and b) by combining a target variable with different context variables i.e., to provide the inputs $(y, x_1)$ and $(y, x_2)$ to the networks as shown in figure 1. $x_1$ and $x_2$ are different but related signals which help create alternative perspectives on $y$. Agreement can only be achieved on an abstract level in the sense that the networks agree on the occurrence of data configurations to which they assign codes that can be translated between the networks using a stable dictionary. In the context of finance, these configurations are often referred to as market regimes. The advantage over a single auto-encoder of all available signals $(y, x_1, x_2)$ is that our set-up allows the AEs to develop independent perspectives which are subsequently aligned. This retains the focus on the target $y$ which would otherwise be dominated by the context variables when minimizing the reconstruction loss.

## 2 RELATED WORK

The de-noising problem is well-known in finance and has traditionally been addressed using linear filter methods (such as moving averages, Bollinger bands, Kalman filtering). In a recent paper Liu & Cheng (2024), mode decomposition and a wavelet-thresholding method are employed to classify stock price movements. Other methods employ a generative model to reproduce known stylized features of market prices e.g., based on Langevin dynamics Wang & Ventre (2024). The paper by Bao et al. (2017) presents stacked autoencoders for hierarchically extracting deep features in stock prices which are used as an input to an LSTM to forecast next day's closing prices. The estimation procedure in this paper is comparatively simpler and consists in a pair (or more) parallel auto-encoders. The novelty here is that the AEs mutually regularize their learned representations.

As stated in the introduction, regularization has been an integral part of the development of autoencoders. Without it, they would merely produce a literal copy of the input without extracting valuable features from data or be able to reduce the dimensionality of its representation. Classic approaches involve $L_1$ and $L_2$ constraints on the network weights or robust training techniques like dropout Baldi & Sadowski (2013) that avoid over-dependence on particular activation patterns. This paper takes an interactive approach which similar in spirit to adversarial regularization Makhzani et al. (2015), Zhao et al. (2018). The adversary sub-network encourages the main network to learn an unbiased representation. This has been shown to yield undesirable side-effects, including unstable gradients and reduced performance on in-domain examples Grand & Belinkov (2019). By contrast, our setting is a collaborative one in which the encoder outputs of two networks are compared given the same input data. Our idea is to reconcile these outputs in terms of the synchronicity of their appearance.

## 3 REGULARIZATION

Autoencoders project the input data on a low dimensional manifold $M$ whose coordinates are given by the codes $z$ in the hidden layer. They are typically designed to be undercomplete, i.e. the dimension of their encoder outputs is much smaller than the input dimension. This prevents them from learning the identity function. Because of the bottleneck, some aspects of the input have to be grouped into larger components from which the input is reconstructed while a literal reproduction is avoided. These components represent the features that define the objects contained in the data. However, the relationship between architectural constraints and simplification is quite loose and is typically determined empirically. In this paper, we propose a scheme in which the simplification is the result of a negotiation among two networks.

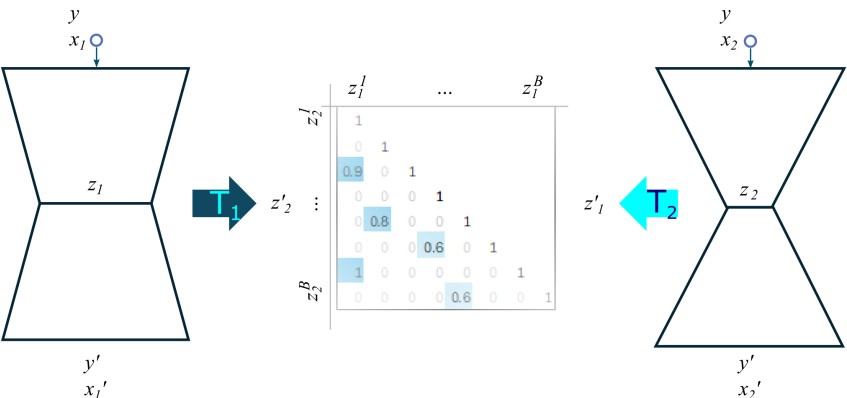

Figure 1: Two AEs communicate with each other (via translators) at the level of the encoding layer while trying to reproduce input data.

The networks only have weak bottlenecks and may produce over-fitted predictions. The key point is that the models are different in the sense that predictions are obtained by decoding two different sets of codes $z$. If the networks assign codes to different perceived regularities in the data they can only agree on their (posterior) distribution $p(z|x)$ if the different codes repeatedly co-occur. If a code has been assigned to noise, the probability of its repeated co-occurrence with a code generated by the other network is very low. On codes for which an agreement on $p(z|x)$ is possible on the other hand must refer to a true regularities in the data.

The more the two networks differ the more effective they are as mutual regularizers. To achieve a different encoding behavior the networks can be of different complexity (e.g., by the number and width of layers). Also, independent noise sources $n_1$, $n_2$ can be added (as shown in figure 1) whose role is to encourage the networks to search in different directions. Notice that the noise is applied at the input and output during the training of the AE (unlike the denoising setup). (Very loosely speaking,) it can be seen as an additional source of entropy which helps explore the space of possible code words. The objective is to find a common denominator with the other network, which may lie outside of the code regions explored by the networks individually. By mutual agreement we expect not only the noise to be rejected, but also the predictions to be more standardized as they are reproductions of the input data which are decoded from an agreed-upon posterior.

## 3.1 TRAINING SETUP

The training algorithm simultaneously improves the data fit and the mutual understanding on how to achieve it. Training is organized as a conversation among two AEs. The networks take turns at talk and either speak (S) or listen (L) to the other network. The S/L phases are asynchronous as in natural conversations. Translation layers $T_1$ and $T_2$ are placed between the two AEs to convert to and from the encoder outputs $z_1$ and $z_2$. In view of the desired dissimilarity the encoding layers may have different output dimensions e.g., $\dim z_1 > \dim z_2$ (without loss of generality). The loop starts with a speak action (S) by one of the networks, say $AE_1$. Utterances are produced by sampling from a multivariate Gaussian with mean $\mu_1(x)$ using the familiar re-parametrization trick

$$z_1 = \mu_1(x) + \sigma\,\varepsilon \quad \varepsilon \sim \mathcal{N}(0,1) \tag{1}$$

where $\mu_1(x)$ is learned by the encoder and $\sigma > 0$ is fixed for simplicity. We write $z_1 \sim q_1(z_1|x)$ where $q_1(z_1|x)$ is the a posteriori distribution of codes $z_1$ after seeing the data $x$. It depends on the network parameters of $AE_1$. A deterministic decoder maps $z_1$ to the network output $x_1'$ which represents the prediction of the input data $x$. During (S), no parameters are updated. While $AE_1$ speaks, $AE_2$ listens.

TRAINING THE AUTOENCODER

Training only occurs during the listening phase. The following training procedure is described from the perspective of $AE_2$ but equally applies to $AE_1$ once it listens (with the roles of all variables switched accordingly). By means of the translation layer $T_1$, $AE_2$ obtains an estimate $z_2'$ of its own code $z_2$ as produced by $AE_1$:

$$T_1 : \; z_2' = g(z_1, \bar{z}_1, \bar{z}_2) \quad z_1 \sim q_1(\cdot|x) \tag{2}$$

where $\bar{z}_1$, $\bar{z}_2$ are pairs of codes obtained from a previous training round (see next section). This transformation gives $z_2' \sim p_1(\cdot)$ where $p_1(\cdot)$ is no longer a posterior distribution from the perspective of $AE_2$ but represents data-independent noise. It depends on the parameters of $AE_1$ which, however, are not updated while $AE_1$ speaks. Predictions $x'$ of $x$ will be generated by sampling codes $z_2 \sim q_2(\cdot|x)$ from the posterior distribution of $AE_2$. We assume that there exists a decoder $p_2(x|z_2)$ depending on $AE_2$'s parameters that describes the joint distribution $p_2(x, z_2) = p_2(x|z_2)p_1(z_2)$ well. This means that

$$\mathbb{E}_{z_2 \sim q_2(\cdot|x)} \underbrace{\left[\frac{p_2(x, z_2)}{q_2(z_2|x)}\right]}_{\xi(z)}$$

is an unbiased estimator of the data-likelihood $p_2(x)$. From this we obtain $\ln p_2(x) = \ln \mathbb{E}_{z_2 \sim q_2(\cdot|x)} \xi(z) \geq \mathbb{E}_{z_2 \sim q_2(\cdot|x)} \ln \xi(z) :- L_2(x, q_2)$ by Jensen's inequality, where $L_2$ is the ELBO associated with $AE_2$. $L_2$ may be re-written as

$$L_2(x, q_2) = \mathbb{E}_{z_2 \sim q_2(\cdot|x)} \left[\ln p_2(x|z_2) - D_{KL}(q_2(\cdot|x)\|p_1(\cdot)\right] \tag{3}$$

We measure $\ln p_2(x|z_2)$ with the help of isotropic Gaussians $\mathcal{N}(x_2', \mathrm{diag}\, \sigma^2)$ centered at predictions $x_2'(z_2)$ and using the same variance $\sigma^2$ as in equation (1) so the term becomes proportional to $-1/\sigma^2 \|x - x_2'\|^2$. The KL divergence between the approximate posterior $q_2(z_2|x)$ and the data independent prior $p_1(z_2)$ is the divergence between two Gaussians

$$D_{KL}(q_2\|p_1) = 0.5 \left[\ln \frac{|\Sigma_{p_1}|}{|\Sigma_{q_2}|} - \mathrm{dim}(z_2) + (\mu_{q_2} - \mu_{p_1})^T \Sigma_{p_1}^{-1} (\mu_{q_2} - \mu_{p_1}) + \mathrm{tr}[\Sigma_{p_1}^{-1} \Sigma_{q_2}]\right] \tag{4}$$

We let

$$z_2 = \mu_2(x) + \sigma\, \varepsilon \quad \varepsilon \sim \mathcal{N}(0, 1) \tag{5}$$

using the same scaling factor $\sigma$ as in (1) so the parameter dependent part of $D_{KL}(q_2\|p_1)$ becomes proportional to $1/\sigma^2 \|\mu_{q_2} - \mu_{p_1}\|^2$. In summary, we obtain:

$$L_2(x, q_2) = -\mathbb{E}_{z_2 \sim q_2(\cdot|x)} \left[\|x - x_2'(z_2)\|^2 + \|\mu_{q_2} - \mu_{p_1}\|^2\right] \tag{6}$$

which is maximized during the listening (L) phase of $AE_2$. The presence of $p_1(\cdot)$ acts as a (regularization) obstacle that prevents the encoder from fitting an arbitrarily complex posterior $q_2(\cdot|x)$

THE TRANSLATION LAYER

$g(\cdot)$ in equation (2) is realized by a non-trainable attention layer which associates $z_1$ with $z_2$ by means of key-value pairs $\bar{z}_1$, $\bar{z}_2$ obtained from a previous training loop (and initialized by training the two AEs separately during the first epoch). A kernel function

$$k(z_1, \bar{z}_1) = \exp\left[-\|z_1 - \bar{z}_1\|^2\right] \tag{7}$$

allows us to retrieve the closest match between any given $z_1$ and the set $\bar{z}_1$. Every such matching key has a value $z_2 \in \{\bar{z}_2\}$ associated with it. This yields a per-sample translation among representations and is in contrast to learning a smooth functional relationship between $z_2$ and $z_1$ which would lack the sharpness needed for regularization. This is illustrated by the dashed line in figure 2. The shape of the line is dominated by the three data patches in the lower, middle and upper part of the diagram. Bearing in mind that both $\mathrm{dim}\, z_1$ and $\mathrm{dim}\, z_2 \gg 1$, the natural situation is that there will be less populated areas or "gaps" which are only occupied by a few samples. These samples do not contribute materially to the loss function of the hypothetical model that is fitted to learn $g(\cdot)$ and are therefore ignored. However, in the diagram, they form an equivalence class (versions of the letter "a") which are all mapped to the same code (calligraphic a) as $AE_2$ perceives them as equal.

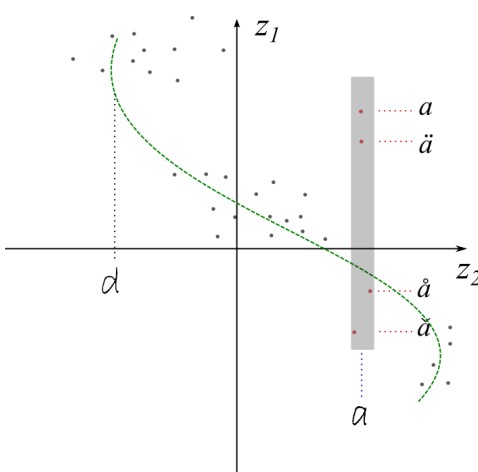

Figure 2: The translator

This is precisely the information that will regularize $AE_1$ and must therefore be retained rather than "averaged" out. In other words, the translator must provide a per-sample association between $z_1^t$ and $z_2^t$ for every data point $x_t$, $t = 1, \ldots, T$. The same is true for

$$\text{T}_2: \ z_1' = h(z_2, \bar{z}_1, \bar{z}_2) \quad z_2 \sim q_2(\cdot|x) \tag{8}$$

which is the analog of equation (2) in reverse direction. $AE_2$ will use $\text{T}_2$ in the subsequent speak (S) phase to produce $z_1'$ which $AE_1$ obtains as a regularization input. $AE_1$ then assumes the role of the listener (L) and the same update rules (with swapped variables) as described a above are applied. The training procedure is summarized in figure 3: standard training of separate networks only contains "speak" phases while mutual training features turn-taking among the networks.

The right hand side of the figure presents a typical evolution of the training error. The need to incorporate translated codes received from the other network introduces a non-monotonicity which has to be overcome by aligning the codes. The sigmoid activation functions in the encoding layer have a discretization effect on the codes that need to be aligned. By consequence, the error landscape is more rugged than the one obtained with standard training. The experiments behind these data are further described in the next section.

It should be noted additional conversation partners may be added which communicate with either one or more existing ones. For example we may introduce $AE_3$ which communicates with $AE_2$ by providing translations $\text{T}_3$ of $z_3$ to $z_2$. The loss function (6) is then modified as

$$L_2(x, q_2) = -\mathbb{E}_{z_2 \sim q_2(\cdot|x)} \left[ \|x - x_2'(z_2)\|^2 + \|\mu_{q_2} - \mu_{p_o}\|^2 \right] \tag{9}$$

where $p_o$ alternates between $o = 1$ and $o = 3$ depending how the turns at talk are organized.

## 4 REGULARITIES IN FINANCIAL TIME-SERIES

Financial markets are in general very efficient at converting new information into prices of traded securities. This means that changes in prices are essentially driven by the random arrival of new information. As a consequence, 'noise' dominates the evolution of financial time-series. Regularities arise when market participants irrationally disregard available information or have difficulties interpreting it. The regularity represents a market inefficiency which becomes a source of risk-free return (alpha) if exploited by a trading strategy. For example, the prices of two related commodities may temporarily be out of sync which creates a price spread that eventually has to disappear because the more expensive commodity may be substituted by the cheaper one creating demand for the latter. Most systematic trading strategies available today are handcrafted, i.e. they start with an inefficiency which is underpinned by an economic theory, e.g. behavioral finance. It is reasonable to expect that many more regularities exist in financial times years which however are hidden behind high levels of noise. An auto-encoder with strong denoising capabilities opens the door to a

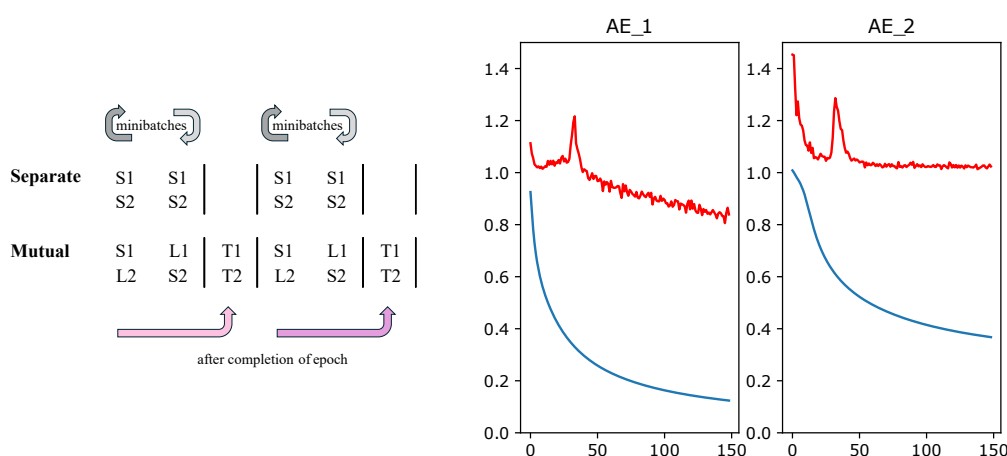

Figure 3: Separate (in blue) vs. mutual learning (in red) of two AEs

potentially large strategies space and may help discover new sources of alpha. This is illustrated, in the following experiments. It must be noted that we use the profitability of a trading strategy as an indication of whether the denoising is successful or not. This is because "ground truth" i.e., the low-noise skeleton inside the time-series, cannot be observed in practice.

The regularities we seek to discover in time-series consist of longitudinal and cross-sectional ones. We employ a convolutional auto-encoder to capture them, see figure 4. The bottleneck layer (output of the encoder) flattens the time-step dimension (i.e., the dimension representing the rolling window over data used in the convolutions) and transforms the result via a dense layer into a low-dimensional code making the overall network under-complete. The network is trained using an ADAM optimizer with learning rate 0.01. The data consists of weekly macro and price data starting in the mid 1980's which gives about 2000 samples. This relatively small data set corresponds to the typical information available to traders operating on the mid-to long-term horizon. We subdivide the dataset into 10 mini-batches and train for 150 batches. The translator is implemented as a simple dense network with a single, wide, hidden layer. As mentioned above, the objective of the translator is not simplification but to implement a one-to-one dictionary between the codes generated by the partner networks. Again, ADAM is used to optimize with step-size 0.1 over 10 epochs. All training phases (see figure 3) are incremental, i.e. the network weights obtained in a given epoch form the initial weights at which the next training epoch starts.

In the experiments, we will study how predictions obtained from an autoencoder with this architectural constraint improve by adding the mutual regularization constraint. In particular we are interested in emergent patterns among the context variables once the AEs interact with each other. The experimental setup consists in a target variable $y$ being combined with different context variables $x_i$, $i = 1, 2, \ldots, N$. The variables refer financial time-series where $y$ corresponds to the returns of a target market (to be traded) and $x_i$ to environment (e.g., macroeconomic) data. The returns of the target market are shifted forward $h$ time-steps during the training phase. The motivation behind this set-up is that if a relationship between today's environment and the $h$-steps forward returns of a target market is found, then $x_i$ provides a trading signal for $y$. In such a case, $h$ corresponds to the investment horizon of the trading strategy based on $x_i$.

For every pair $(y, x_i)$, an autoencoder $AE_i$ is trained which produces a denoised version $(y', x_i')$ of the input. The idea is that after denoising the relationship between $x_i$ and $y$ is more stable and apparent. The objective is to learn "typical" evolutions of $x_i$ that precede market up– or down–moves. These typical evolutions are obtained by clustering (e.g., by standard K-means) the denoised $x_i'$ conditional on $y > 0$ (up-market) or $y < 0$ (down-market). The evolutions are truncated at a length that corresponds to the "depth" dimension of the convolution layers (32 timesteps in this case, see figure 4). The clustered patterns form a library of signals that can be used out-of-sample

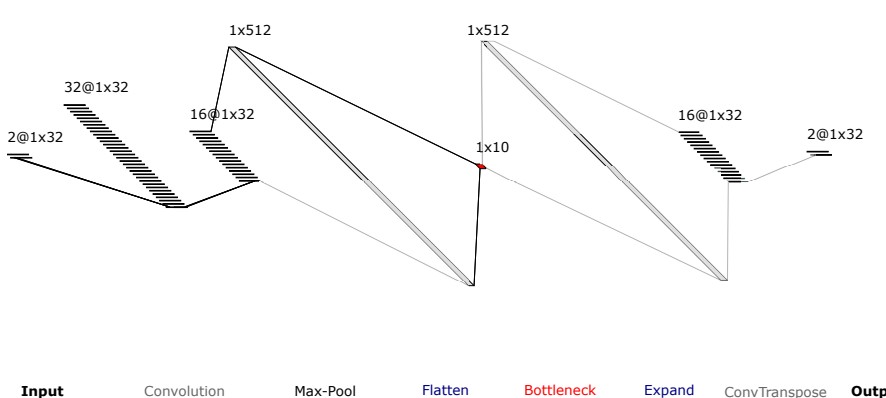

Figure 4: Architecture of one AE: encoding and decoding stages employ (1D) convolutional layers and are linked through a bottleneck layer in which the "time-step" dimension of the convolution output is flattened. Heterogeneity among different AEs is obtained by varying the number of the input convolution channels between 16 and 32.

to decide whether to build a long or short position in the target market. Since $y$ is shifted forward, out-of-sample data cannot be auto-encoded, so only the original (noisy) $x_i$ can be compared to the library. However, as every pattern corresponds to a (truncated) time-series our feature space is fairly high dimensional which means that we can robustly determine the relative closeness of noisy signals towards the up– or down–patterns in the library.

Indeed, we define a trading position in terms of relative distances as

$$\theta_i = d_{iu}^{-1}/(d_{iu}^{-1} + d_{id}^{-1}) \tag{10}$$

where $d_{iu} = \sum_{\xi_u} \mathrm{MSE}(x_i, \xi_u)$ and $\xi_u$ is a library entry corresponding to an up-market and MSE stands for "mean squared error". $d_{id}$ refers to the distance to all down-market library patterns. This definition of $\theta_i$ corresponds to a simple (directional) trading strategy which directly exploits relationships extracted from denoised configurations $(y, x_i)$. The strategy can be refined by combining libraries obtained from multiple AEs. At this stage, finding common ground among the AEs is an important prerequisite as it not only leads to more abstract reconstructions of the target–context pairs but also allows us to mix contexts. When mixing signals it is practical to proceed in a pairwise fashion. This provides an interesting use-case for mutual regularization as introduced in this paper.

The target variable ($y$) is the S&P 500 (total return USD). The context variables ($x_i$) used in figures 4-6 are: Y10 = 10-Year Treasury Yield, CAPE = Cyclically Adjusted Price/Earnings Ratio, NYF = New York Fed Economic Activity Index, MG = US Corporate Margins (YoY), Y02 = 2-Year Treasury Yield/ short-term rates, STP = Steepness of the Treasury Yield Curve, M2 = Money Supply (YoY). In figure 5, we study configurations of context variables namely Y10 (used by AE$_1$) and CAPE (used by AE$_2$) while the target market (S&P 500) is color coded as up (red) or down (blue). For illustration purposes homogeneous up or down regions are red or blue shaded. We find that the separate reconstructions achieve some sorting relative to the original data. The output of AEs with mutual regularization, in turn, reveal that up and down phases can be very clearly separated in Y10-CAPE space. It should be noted that all three representations yield the same reconstruction of the target variable but differ in the location its up/down values in the space spanned by context variables. Strong denoising is also apparent as the data occupies a smaller region in Y10-CAPE space.

Figure 6 illustrates how the sorting of context variables allows for the extraction of characteristic profiles associated with up or down markets. The profiles consist of sequences of length 24 (weeks) and are extracted by clustering the reconstructed (i.e., denoised) context variables conditional on positive/ negative *reconstructed* future returns of the target market. The profile form references against which new context data will be compared. Notice that the new data cannot be processed by the AE as the pair $(y^{t+h}, x_i^t)$ contains future (unknown) returns $y^{t+h}$ which are only available if $t$ is $h$ steps in the past ($h$ is the investment horizon). The role of denoising (by the AE) is to create stable and sufficiently different reference profiles against which noisy (unprocessed) data can be compared

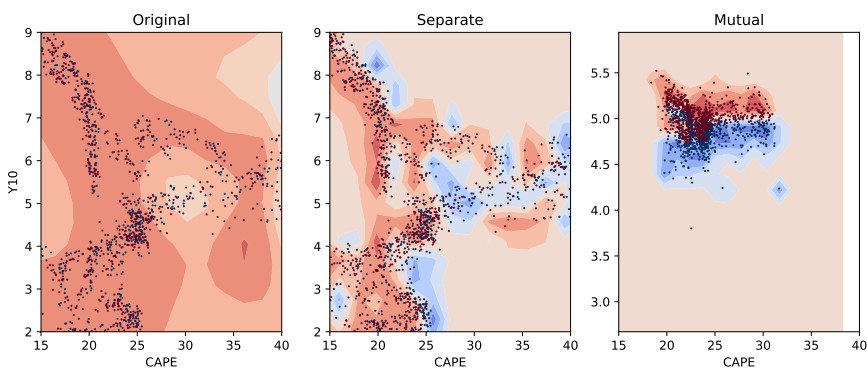

Figure 5: Reconstructions of context variables $x_1$ =Y10, $x_2$ =CAPE; left: literal reconstruction (no regularization), middle: separate training of AEs), right: training of AEs with mutual regularization. Up/ down values of the target variable $y$ =S&P 500 are color coded as red and blue.

out-of-sample. If trading positions in the underlying market are entered according to (10) a payout profile emerges with interesting diversifying properties relative to the market in that it avoids the 2020 and 2022 draw-downs induced by the onset of the pandemic or the rate hike cycle respectively.

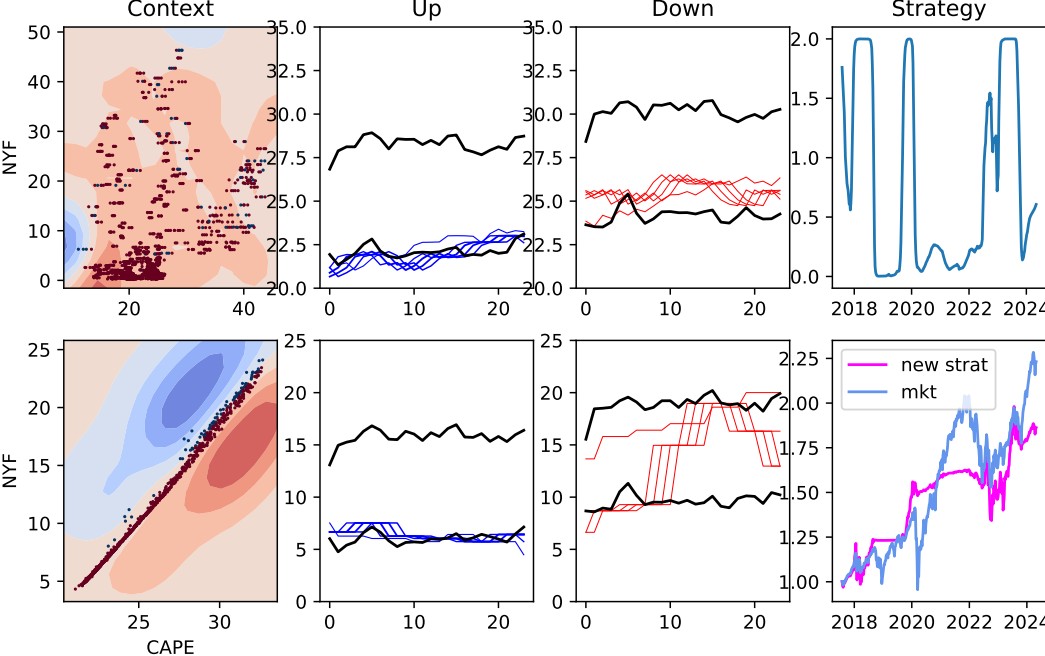

Figure 6: Strategy discovery from denoising environment variables: characteristic profiles are extracted to which new data is compared to define a trading position in the market for which a context representation has been found.

By forming random pairs from the available context variables we can efficiently create other strategies which derive trading decisions from different configurations of context variables. Figure 7 provides examples of strategies obtained by combining other pairs of contexts. We see that the

resulting trading positions and P/L (profit and loss) time-series are quite different and respond to different macro-events. While CAPE/NYF and CAPE/MG respond to fundamental economic activity and margin growth (compared to what is priced into the valuation multiple) the three other strategies introduce the vantage point of rates (Y02, or RR) as well as money supply (M2). Interest rates determine how future cash-flows received from owning the stocks should be discounted and thereby impact the valuation multiple. The middle strategy PE/Y02 stayed out of the market after the first rate hikes occured before 2020. it is also much less sensitive to rates when the hiking resumed after the pandemic. This is very different from strategy STP/M2 which recognizes that the enormous expansion of money supply during the pandemic eventually inflates prices including those of stocks. We see that despite the abstract nature of the AE itself, its output can be understood and interpreted especially when forming pairs of AEs that coordinate their learning through mutual regularization.

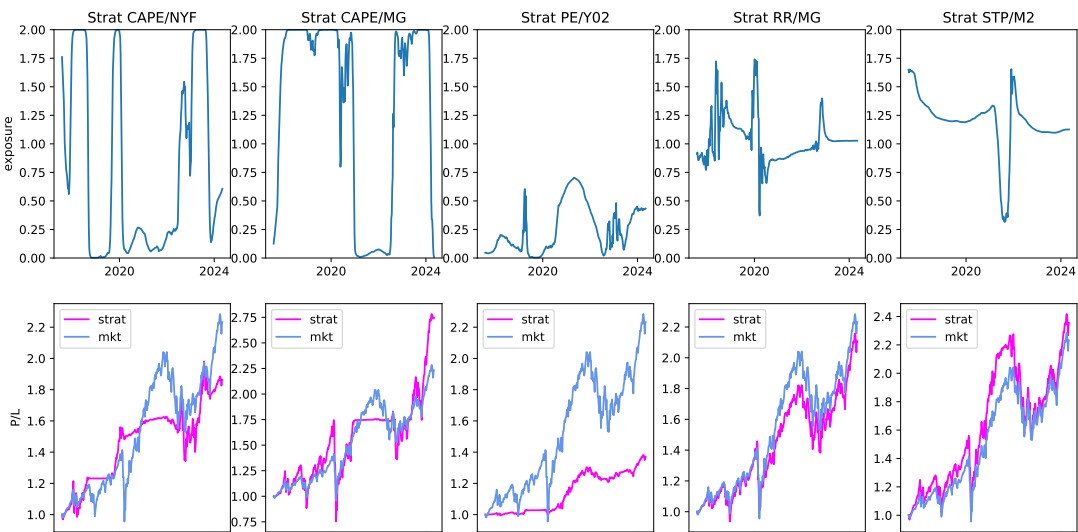

Figure 7: Overview of trading strategies obtained by denoising pairs of context variables together with a (shared) target variable.

## 5 CONCLUSION

The paper discusses collaborative regularization where two networks compare notes in the form of encodings generated upon seeing the same input data. A useful analogy is to think of the codes as words in different languages denoting the same object. The expressions might have different connotations which however have to be neglected in order make an agreement on their usage possible. This implies a degree of *standardization* in what can be predicted using agreed-upon codes. The codes correspond to recurring features in the input data, which are therefore also the defining elements of the objects contained in the data. Every input data point gives rise to a code and if this code is of the "agreed-upon" type, it means that the data point is *representative* of the object or pattern to be identified. This conclusion can be reached in an entirely unsupervised fashion. In some sense, the networks mutually supervise themselves by trying to match their use of encodings. Agreed-upon codes define equivalence classes of inputs (one for each code).

The idea opens up a to a vast strategy space which has yet to unfold entirely as more and more stable relationships are identified and traded. This will contribute to making financial markets more *complete* by providing liquidity and putting a price on more assets in more states of the world. Many active strategies are based on well-known (stylized) market inefficiencies or they operate within

significant constraints (e.g., within a tracking error limit vs. a benchmark). If all market participants search along a similar dimension, alpha capture becomes a zero-sum game. A market with more heterogeneous trade positions can make all participants better off –according to their own criteria– without having to make someone else worse off (Pareto optimality with differential preferences).

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
