# OpenReview forum: "Strong denoising of financial time-series"
_ICLR.cc/2025/Conference — Submitted to ICLR 2025_

### Official Review · Reviewer_ioke · 2024-11-02

**Soundness:** 1
**Presentation:** 1
**Contribution:** 2
**Rating:** 3
**Confidence:** 3

**Summary:**

This paper proposes a  method using autoencoders to improve the signal-to-noise ratio in financial time-series data. The approach involves using two AEs that collaborate through a "conversation," alternating between producing predictions and reconciling differences to enhance data representation. The goal is to extract meaningful features and denoise financial data, which has a  low signal-to-noise ratio. Experiments suggest that this  regularization approach supports finding market regularities and helps the development of new trading strategies.

**Strengths:**

**Originality**: The paper introduces a method for enhancing financial data representations by using a regularization between two auto-encoders, hereby I think the conversational aspect is novel.

**Clarity**: The methodology, especially the "speaking and listening" phases of the auto-encoders, is well explained

**Significance**: In general i think the work could have significant implications for more robust trading strategies. The proposed approach could lead to new insights in financial forecasting by enhancing the clarity of underlying market patterns.

**Weaknesses:**

My background is not in the context of financial time-series, however I really struggled understanding what the authors try to describe with their provided figures, e.g., what is a strat and mkt? In the following i outline some areas which might allow for improvements

**Figure Quality**:
- In general, the figures in the manuscript need a significant revision
- Figure 2 uses hand-drawn letters, which is not appropriate for a venue like ICLR.
- Axis labels are overlapping, and there is no legend for the blue, red, or black series in Figure 6.
- Figure 4 does not provide substantial value; a table would suffice. Additionally, the use of color in the layer names is not explained, making it misleading.
- Figure 3 lacks axis labels, which makes interpretation difficult.

**Reproducibility**: The absence of a code or detailed procedural steps limits the reproducibility of the experiments. Providing a link to code or pseudocode would help mitigate this issue.

**Experimental Scope**: The experimental dataset used (around 2000 samples) is relatively small, especially in the context of financial data. Broader testing with larger datasets is needed to substantiate the claims made in the paper. Additionally, I think qualitative evaluations alone are insufficient in this case. A more thorough quantitative evaluation is required. Showing improvement over older/other denoising methods (e.g., traditional filters or more advanced nn approaches) would provide stronger support for the novelty and effectiveness of the proposed method.

Method Justification: The design choices are not well-supported by ablation studies or references to related literature. It would be helpful to mention similar concepts like Siamese Networks (e.g., Dong et al.) and mutual information alignment by Lee et al.

Related Work: Basing on the pure number of references and the amount of demand in financial forecasting (as the authors motivated), i do not believe the related work is nowhere near exhaustive.

### References:

Lee et al., Soft Contrastive Learning for Time Series (ICLR 2024)

Dong et al., TimeSiam: A Pre-Training Framework for Siamese Time-Series Modeling (2024)

**Questions:**

1. Could you clarify why you think the test on this relatively small dataset provides any signifiance?

2. How would the proposed method perform against similar approaches? Are there particular benefits that could be highlighted more explicitly?

3. Is there a plan to release the code for reproducibility?

4. What exactly do we see in Figure 3? Validation Loss?

---

> ### Author Response · Authors · 2024-11-25
>
> Weaknesses:
> - strat = (trading) strategy, mkt = (underlying) market
> - Figure 2 is actually not hand-drawn but a special font used to denote equivalence classes (can of course be changed)
> - thank you for the detailed improvement suggestions for the figures
>
> - experiments: unless we investigate high-frequency trading, the majority of financial time-series datasets is indeed small. This is because most of the macro-variables (used as context variables $x_i$ in the paper) are only available on a monthly or quarterly frequency. The scope of the experimental study will be expanded by considering other underlying market/ context combinations but each one of these experiments will be on a small dataset.
>
> - method justification: the experiments were conducted using a minimal set-up with two shallow auto-encoders. The scope for conducting an ablation study is therefore limited.
> - other denoising methods: yes, a comparison of the experiments against older methods would strengthen my case. Classic approaches assume that the noise is white/ has small amplitude/ can be separated from the signal in frequency domain. None of these assumptions is true when exploiting financial data.
> - thank you for pointing out additional references. I am aware of these papers but did not include them as NCE is a method of self-supervision (while this paper is about mutual regularization) and Siamese Networks are the exact opposite of the proposed method (in that the networks are identical while the AEs in this paper are *designed to be different* to exploit two "perspectives" for denoising task). I will include the references to contrast the proposed method against them.
>
> The code will be made available upon acceptance of the paper

---

> > ### Comment · Reviewer_ioke · 2024-11-28
> >
> > Thank you for these clarifications. You addressed some of my concerns.
> >
> > I maintain my score.

---

### Official Review · Reviewer_W61n · 2024-11-02

**Soundness:** 2
**Presentation:** 1
**Contribution:** 1
**Rating:** 3
**Confidence:** 3

**Summary:**

This paper addresses a method for modeling time-dependent signal data, such as financial time series. The method employs two autoencoders with different network architectures, assuming they learn different features. The emprical results are demonstrated mainly on S&P500 data.

**Strengths:**

This work presents a new method to tackle the problem of time series prediction.

**Weaknesses:**

- I find some of the components of this paper not well-defined. Terms "listen" and "speak" are fine to use as long as they are well defined. I understand the intuition but cant verify if what I understand is correct.

- Some components of this paper are not well-defined. Terms like "listen" and "speak" are acceptable as long as they are clearly defined. While I understand the intuition, I cannot verify if my understanding is correct.

- There is a lack of quantitative results (I am not familiar with financial benchmarks). The authors present strategies obtained by the model in Figure 7, which, from my perspective, seem to mostly predict following the present (i.e., the prediction is a one-day shift from the ground truth). This observation might be incorrect, but I cannot verify it. Therefore, I am missing a few metrics that indicate the model's quality, such as the profit over 1 or 5 years compared to simply staying invested in the S&P 500.

- The paper includes a lot of details that are obvious at this point, such as reparameterization trick derivations.

- I see no reason to use this method exclusively on financial data. While financial data is a challenging domain, it would be beneficial to see this method applied to more popular modalities or benchmarks.

- There is no comparison to prior works.

I could not find any code attached to this work. I believe that providing code that reproduces the paper's results adds significant value to any research work.

**Questions:**

What is the difficulty of training such method? It seems like the models may collapse.

---

> ### Author Response · Authors · 2024-11-25
>
> Strengths:
> - the paper tackles the problem of denoising, not prediction. Trading (based on implicit prediction) is only the downstream task used to evaluate the robustness of the denoising.
>
> Weaknesses:
> - the terms "speak" and "listen" are defined in section 3.1 as part of the description of the training setup. I will provide a more concise definition to improve readability.
> - the S&P500 is arguably the most widely used benchmark for introducing and testing new trading ideas - which in our case serve as downstream task for testing the signal/noise separation achieved by our method. This needs to be compared against trading the same signals but using a different de-noising method (not against holding the outright S&P500 without trading).
> - it is true that the sentence before equation 3 could have been omitted. I thought it improves clarity of exposition. Notice that the reparametrization trick is repurposed to introduce signals from the partner AE.
> -  You are right. Financial data has the unique property that the usual assumptions like amplitude or frequency separation of signals vs. noise do not hold. A similar situation may be created by treating all 5s in MNIST digits as noisy versions of one standard 5 which would be the desired result after denoising (similarly for the other digits)
> - Prior work: true - I will first have to find similar work on a challenging denoising task (i.e., not the usual denoising setup where noise is white or some of its properties are known)
>
> Code will be made available upon acceptance of the paper
>
> Question:
> - Why do you think the model might collapse? If the regularization constraint (the second term in equation 6) becomes dominant, the model ceases to reconstruct well but this can be easily remedied with a smaller weight on the regularization term (like with any regularization).

---

> > ### Comment · Reviewer_W61n · 2024-11-26
> >
> > I thank the authors for the answers to my questions, however, I think that this work could benefit from more clarifications and experiments. Hence, I keep my initial score.

---

### Official Review · Reviewer_CUyQ · 2024-11-04

**Soundness:** 1
**Presentation:** 2
**Contribution:** 1
**Rating:** 3
**Confidence:** 2

**Summary:**

This paper presents a novel method for enhancing the signal-to-noise ratio in financial time-series data through a unique application of auto-encoders (AEs). Unlike traditional methods, which often rely on regularization by complexity penalties, this approach uses a dual-AE system where two AEs iteratively refine their reconstructions through a conversational process, leading to mutual regularization. By enforcing agreement between the networks, only genuine data regularities are captured, reducing noise. The authors validate the approach through experiments showing the effectiveness of this denoising technique in creating interpretable signals, benefiting systematic trading strategies.

**Strengths:**

The paper introduces an innovative regularization mechanism, where two AEs mutually regularize each other’s representations. This approach is novel and avoids the drawbacks of traditional regularization methods.

The methodology is detailed, with a clear description of the interaction between the AEs and their translation layers. Additionally, empirical results on financial time-series demonstrate how the denoised signals can be applied to real-world trading scenarios.

The paper is generally well-written, especially in the sections explaining the conversational setup between the AEs and how mutual agreement results in effective noise reduction.

The approach could significantly impact finance by improving time-series predictability and revealing hidden data patterns, contributing to more robust trading strategies. The mutual regularization approach could also have broader applications beyond finance, potentially in any domain involving high-noise data.

**Weaknesses:**

The dual-AE setup may introduce additional **computational overhead**, especially with high-dimensional data. The paper does not discuss scalability or computational trade-offs.

The paper lacks **quantitative metrics** to objectively evaluate the performance of the denoising method. Only qualitative results are presented. Additionally, there is no **comparison with other state-of-the-art** denoising techniques, such as those based on wavelet-thresholding or stacked autoencoders. Including these would provide a clearer understanding of how the proposed method stands relative to existing approaches in terms of accuracy and denoising effectiveness.

The authors do not provide the **code** implementing the method presented in the paper, making it difficult to reproduce the results.

**Questions:**

Could the authors provide a complexity analysis or benchmarks comparing the proposed dual-AE approach with single AE models? Understanding the trade-offs in terms of training time or computational resources would be valuable.

The model employs several parameters (e.g., layer configurations, mutual regularization strength). Could the authors discuss the sensitivity of the model’s performance to these parameters? It would be helpful to know which parameters have the most impact on denoising effectiveness.
To strengthen the evaluation, it would be beneficial to include quantitative metrics to objectively assess the model’s denoising effectiveness.

Additionally, comparing the proposed approach with other established denoising techniques (e.g., wavelet thresholding, stacked autoencoders) would contextualize its performance and help highlight any unique strengths.

---

> ### Author Response · Authors · 2024-11-25
>
> Judging from the very accurate summary, the main idea of the paper seems to have come across.
>
> Weaknesses:
> - Computational overhead was not investigated and it is true that every network that is included into the "conversation" will have to be trained (so the method scales linearly with the number of regularizing networks used). However, if the same number of perspectives (i.e., auxiliary signals $x_i$) had to be learned by a single network, it would have to be larger. The net effect may even be positive because the network are trained to align on their perspectives (rather than learn complicated relationships among them).
> -  The experiments were conducted using a minimal set-up with two shallow auto-encoders. The most important parameter in this set up is the strength of the regularization which has been set to the maximum value before the model breaks down (zero output).
> -  State of the art denoising techniques make assumptions on how the signal vs. noise can be separated (noise is often assumed to have smaller amplitude and be of higher frequency). It is true that the presented method should outperform on the given trading task (which is a necessary downstream objective since any "ground truth" denoised data is not available). Financial data is clearly different from the ECG or weather data which are usually used as benchmarks.

---

### Official Review · Reviewer_XDZm · 2024-11-04

**Soundness:** 2
**Presentation:** 1
**Contribution:** 2
**Rating:** 5
**Confidence:** 2

**Summary:**

This paper proposes a method for learning denoised representations of financial time series. They use distinct autoencoders (AEs) that mutually regularize each other’s learned representations through a collaborative process. To learn these representations, the authors propose training AEs to reconstruct distinct input features corresponding to the same underlying time series (e.g., 10-year treasury yield and cyclically adjusted price/earnings ratio), where the decoder of one AE takes as input the “translated” hidden states of the other AE. This translation is done via an attention layer that associates the two hidden states together via high-similarity. The authors validate their approach with  empirical results showing that their method effectively identifies profitable trading strategies based on denoised relationships between market returns and various macroeconomic indicators.

**Strengths:**

**Novel Approach to Denoising**
The paper introduces a novel concept of using a "conversational" framework with two distinct AEs and translation layers for mutual regularization. This approach differs from traditional methods and offers a potentially powerful way to filter out noise by seeking agreement between independently trained AEs

**Intriguing Application to Trading Strategy Discovery**
The application of this denoising technique to discover trading strategies is a compelling demonstration of its potential. The idea of identifying "typical" patterns in denoised context variables to predict future market movements is interesting and could have practical implications.

**Focus on Practical Relevance**
The paper acknowledges the challenge of the low signal-to-noise ratio in financial data and attempts to address it with a method that aims to extract real market inefficiencies for profitable trading. This focus on practical relevance and potential impact on financial markets is a strength

**Weaknesses:**

**Limited evaluation**
The empirical results are based on a specific set of context variables and a limited time period. It's unclear how well this method generalizes to other markets, asset classes, or timeframes. More extensive experiments and robustness checks are needed to assess the generalizability of the findings.
* The paper relies heavily on the profitability of the discovered trading strategies as evidence of successful denoising. While this is an interesting application, a more direct and quantitative evaluation of the denoising performance itself is necessary. Comparing the denoised outputs to a benchmark or using metrics specific to time-series denoising would strengthen the claims

**Lack of baselines / comparison to related works**
Similarly, I was unable to track the progress of this work versus prior works discussed in the related work. Did the authors run their evaluation on similar denoising methods?

**Lack of clarity / justification in the method**
It was hard for me to really understand the method and its justifications. While the paper describes the architecture of the AEs and the training process, the implementation and function of the "translator" remain unclear. A more detailed explanation of how this component works, including the choice of kernel function and the impact of its non-trainable nature, would be appreciated. After multiple readings, it was still unclear to me how exactly the two autoencoders could learn regularized representations.
* For example, if AE1 and AE2 were both trained with a "self-reconstruction" term MSE_self and "cross-reconstruction" term MSE_cross, then the presented intuition would have made sense, i.e.,:
  * MSE_self = (AE1(x1) - x1)^2 + (AE2(x2) - x2)^2
  * MSE_cross = (Decoder1(z2) - x1)^2 +  (Decoder2(z1) - x2)^2)
  * AE1(x) = Encoder1(Decoder1(x))  (same for AE2)
  * Encoder1(x1) = z1 (same for Encoder 2)
* But I could not tell if such a setup was being used, and the motivation behind instead using an attention mechanism felt unclear to me.

Similarly, how is y learned?

**Questions:**

**Method Clarity**
Can the authors clarify where $\bar{z}_1$ and $\bar{z}_2$ come from? I was confused by this because their definition was deferred in their initial introduction (L170), but this was not defined in the next section.

How exactly does the attention mechanism with key-value pairs work? What is the rationale behind using the specific kernel function in equation (7)? Why the RBF kernel instead of the usual exponential over dot products? Why is the translator non-trainable?
* Suggestions: To improve clarity, I'd include:
  * a step-by-step explanation of the attention mechanism and its implementation.
  * Justification for the choice of kernel function and any hyperparameters involved.
  * Clarification on why the translator is non-trainable and how this affects its performance.
  * A redrawn visual diagram to illustrate exact mapping between x1, x2, z1, z2, and y


**Experiments**
Is there a way you can quantify the effect of denoising? I wasn't quite sure how to interpret the visuals in Figure 5



**Presentation nits**
What are the x and y-axis in figure 2 correspond to? x-axis is training steps? As nit, would be helpful to label in figure.

---

> ### Author Response · Authors · 2024-11-22
> **Clarifications**
>
> *Lack of clarity*:
> - I like your idea of using self and cross reconstruction terms, but the idea in this paper is not to build a powerful decoder that can use either encoder outputs $z_1$ and $z_2$ to reproduce data but to seek agreement among AEs on related but different data sets. This agreement can only be reached if both AEs learn sufficiently abstract (not literal) representations of the data. Strong denoising means to disregard data features that are unlikely from the perspective of the partner network, a concept I call "mutual regularization" (line 84/ line 134).
> - The paragraph "common understanding requires generality" elaborates this point and also explains the role of $x_1$ and $x_2$ (different context variables) vs. $y$ (common target variable)
>
> *Limited evaluation*:
> - I have conducted experiments on around 40 context variables (mostly US macroeconomic data) vs the S&P500 (as target) of which only a few are reported. I agree that the result should be presented in a more systematic and comprehensive way.
> - Direct evaluation of the denoising performance of the method is impossible as ground truth is unobservable (and no benchmark datasets of denoised financial data exist to the best of my knowledge) so I use a downstream task (trading on the denoised reconstruction) to evaluate performance
>
> *Questions*:
> - $\bar z_1$ and $\bar z_2$ are dictionaries of codes (latents) generated by AE_1 and AE_2. Promised exact definition is indeed missing.
> - the method does not rely on a specific kernel function, RBF seems like a natural choice
> - the translator needs to be non-trainable because translations are used as (regularization) obstacles and should not adapt to improve MSE loss (but act against it)
> -  $x_1, y$ and $x_2, y$ are simply stacked vectors which combine a context variable $x$ with a (common) target variable $y$
>
> Thank you for your improvement tips. I will include them in a revised version of the paper.

---

> > ### Comment · Reviewer_XDZm · 2024-11-28
> >
> > Thanks for the responses! I think the authors presented an interesting idea for denoising, but as discussed the current draft could be polished in its presentation of methods, definitions, and results. I thus maintain my score.

---

### Meta-Review · Area_Chair_dNe5 · 2024-12-23

**Metareview:**

The paper introduces a method for learning denoised representations of financial time series. It leverages dual autoencoders trained on a reconstruction task, where they mutually refine their reconstructions through a collaborative process. The autoencoders are constrained to agree on their reconstructions, capturing only the underlying data regularities and thus reducing noise. The approach is validated through its application to trading strategies based on the filtered series.

The reviewers commend the originality of the proposed approach and its target application. However, they raise several concerns, including the lack of clarity in the technical description, a limited evaluation—particularly with respect to quantitative metrics—and insufficient comparisons with baselines. The authors are encouraged to refine their positioning and enhance their experimental evaluation.

**Additional Comments On Reviewer Discussion:**

The reviewers noted a lack of clarity and weak experimental validation. The rebuttal did not change their opinion.

---

### Decision · Program_Chairs · 2025-01-22

Reject